# Mitochondrial respiratory gene expression is suppressed in many cancers

Ed Reznik[1*†], Qingguo Wang[1†§], Konnor La[1], Nikolaus Schultz[1*‡], Chris Sander[2,3*‡]

[1]Center for Molecular Oncology, Memorial Sloan Kettering Cancer Center, New York, United States; [2]Department of Cell Biology, Harvard Medical School, Boston, United States; [3]cBio Center, Dana-Farber Cancer Institute, Boston, United States

**Abstract** The fundamental metabolic decision of a cell, the balance between respiration and fermentation, rests in part on expression of the mitochondrial genome (mtDNA) and coordination with expression of the nuclear genome (nuDNA). Previously we described mtDNA copy number depletion across many solid tumor types (Reznik et al., 2016). Here, we use orthogonal RNA-sequencing data to quantify mtDNA expression (mtRNA), and report analogously lower expression of mtRNA in tumors (relative to normal tissue) across a majority of cancer types. Several cancers exhibit a trio of mutually consistent evidence suggesting a drop in respiratory activity: depletion of mtDNA copy number, decreases in mtRNA levels, and decreases in expression of nuDNA-encoded respiratory proteins. Intriguingly, a minority of cancer types exhibit a drop in mtDNA expression but an increase in nuDNA expression of respiratory proteins, with unknown implications for respiratory activity. Our results indicate suppression of respiratory gene expression across many cancer types.

*For correspondence: reznike@ mskcc.org (ER); schultz@cbio. mskcc.org (NS); sander.research@ gmail.com (CS)

†Ed Reznik and Qingguo Wang are joint first authors.
‡Nikolaus Schultz and Chris Sander are joint senior authors.

Present address: §College of Computing and Technology, Lipscomb University, Nashville, United States

Competing interests: The authors declare that no competing interests exist.

## Introduction

Are the mitochondria of tumors characteristically different from those in normal human tissues? Several recent reports have described somatic changes to the mitochondrial genome (mtDNA) in tumors, which have expanded our knowledge of cancer genetics as well as basic mitochondrial biology (*Davis et al., 2014*; *Joshi et al., 2015*; *Ju et al., 2014*; *Stewart et al., 2015*). In our own work (*Reznik et al., 2016*), we estimated mtDNA copy number for thousands of tumor samples sequenced by The Cancer Genome Atlas (TCGA) consortium and found that several cancer types appeared depleted of mtDNA relative to adjacent-normal tissue.

A drop in mtDNA copy number can decrease the expression of critical (and necessary) proteins of the electron transport chain (ETC)/ATP synthase. Respirometry experiments indicate that cells typically harbor an excess of mitochondrial respiratory capacity, implying that a modest decrease in ETC protein levels will not impact the basal respiration rate of the cell (*Brand and Nicholls, 2011*). However, a sufficiently large reduction in the levels of proteins encoded by mtDNA (e.g. large or complete mtDNA depletion), will precipitate a drop in the rate of ATP generation via respiration. In these cases, cancer cells can partially compensate for lower respiratory ATP generation by increasing flux through glycolysis (i.e. the Warburg effect).

Thus, there remains a gap between the observation of mtDNA copy number changes and their propagation to a drop in respiration. Linking measurements of mtDNA copy number directly with respiratory flux remains difficult: flux studies of comparable sample size to contemporary genomic investigations are not available. However, one can lay a partial bridge between respiration and mtDNA ploidy by examination of mRNA expression of mtDNA (mtRNA). Mitochondrial DNA serves as the template for the transcription of 13 protein-coding genes, each of which is a critical integral

membrane subunit in the electron transport chain or ATP synthase. A synchronous change (increase or decrease) in both mtDNA copy number and mtRNA expression in tumors compared to normal tissue can therefore be used as complementary evidence for a change in respiratory flux, which can then be evaluated experimentally.

Here, we use RNA-sequencing data of TCGA tumors to expand upon our prior study of mtDNA copy number variation in cancer (*Figure 1*). In part, our aim is to determine if tumor-associated changes in the levels of mtDNA-derived transcripts mirror changes in mtDNA copy number. More broadly, we propose using trio measurements of mtDNA copy number, mRNA transcribed from

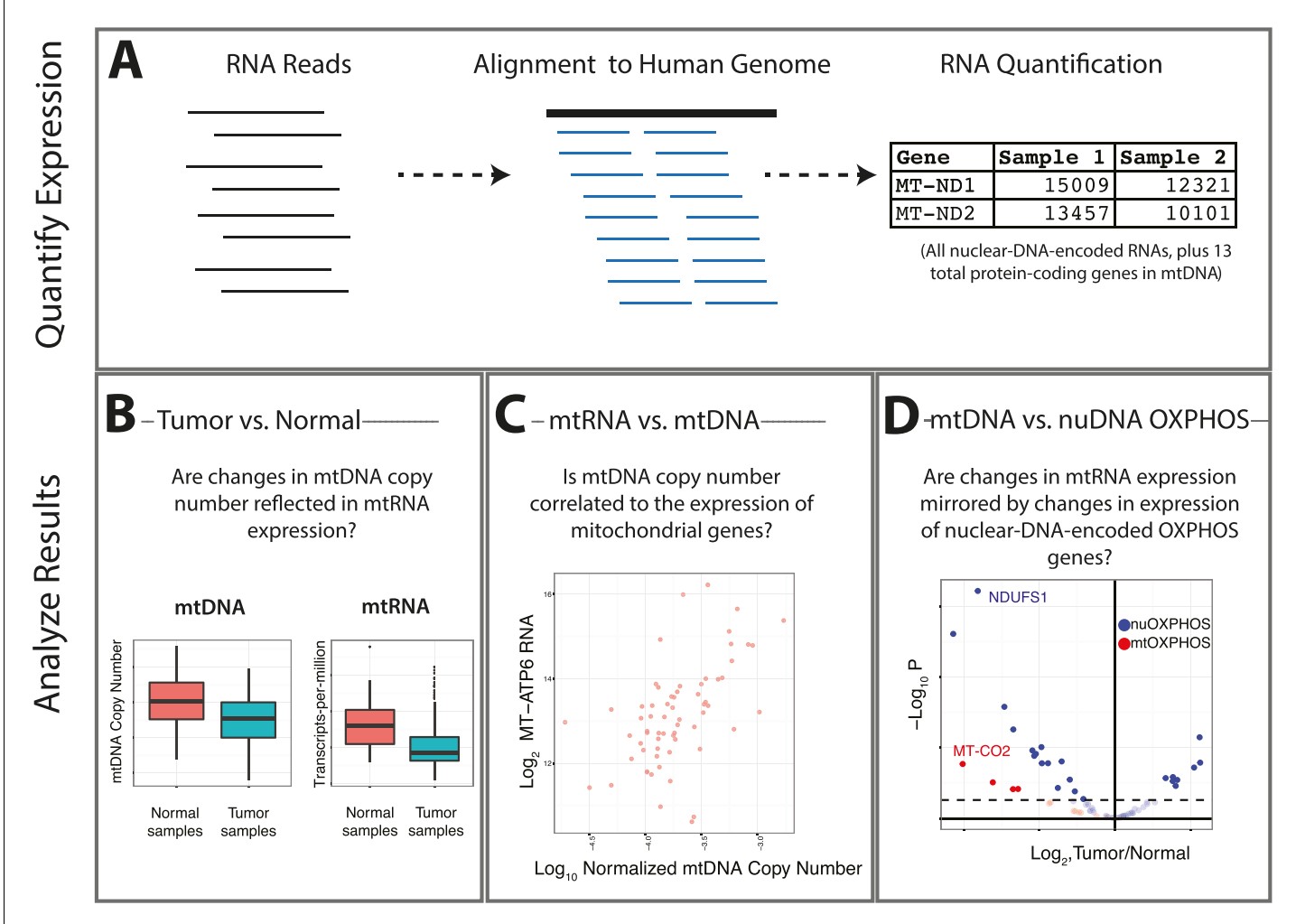

**Figure 1.** Summary of analysis. (**A**) RNA-sequencing reads from the TCGA are aligned, and reads mapping to the mitochondrial genome are retained. (**B**) Changes in the expression of mtRNAs in tumors compared to adjacent-normal tissue are compared to analogous changes in mtDNA copy number. (**C**) Quantitative estimates of the correlation between mtDNA copy number and mtRNA are determined. (**D**) A comparison is made between the tumor vs. normal differential expression of OXPHOS subunits encoded in mtDNA and nuDNA.

The following figure supplements are available for figure 1:

**Figure supplement 1.** For each sample, the $\log_{10}$ ratio of the expression of the 13 mtRNAs (calculated using the sum of their TPM) to the expression of 175 mtDNA pseudogenes (calculated using the sum of their TPMs) was calculated.

**Figure supplement 2.** Estimates of mtRNA abundance and differential expression from RSEM and featureCounts are in good agreement.

**Figure supplement 3.** Comparison of mtRNA expression across different tissues.

mtDNA, and mRNA transcribed from nuclear DNA (nuDNA)-encoded respiratory genes, to gauge the likelihood that a cancer type down-regulates respiration compared to normal tissue. In parallel, these data can be used to dissect the coordination of respiratory transcription, i.e. to compute the correlation between mtDNA and mtRNA levels, and to assess coordination of expression of respiratory genes from the mitochondrial and nuclear genome. As reported in detail below, our results using RNA-Seq data largely corroborate our earlier results using mtDNA copy number (*Reznik et al., 2016*) (with some important exceptions), and additionally offer a lens onto patterns of transcriptional coordination between the mitochondrial and nuclear genomes.

## Results

### Quantifying expression of mtRNAs across cancers

Many reports have examined changes in metabolic gene expression in cancer (e.g. *Gaude and Frezza, 2016*), but no study to date has examined the expression of the 13 mRNAs encoded by the mitochondrial genome. While estimates of gene expression in samples profiled by the TCGA are publically available (e.g. through the Firehose data pipeline at the Broad Institute, http://firebrowse. org), the abundance of the 13 proteins encoded in mtDNA are not reported there. Therefore, to quantify expression from mtDNA, we re-aligned TCGA RNA sequencing data from thousands of the same tumors analyzed in our prior study of mtDNA copy number, and retained for analysis sequencing reads aligning to the mitochondrial chromosome. Estimates of mtRNA expression (in transcripts-per-million, TPM *Wagner et al., 2012*) across 6614 samples profiled is available in *Supplementary file 2*.

As in our prior study, we tested whether mitochondrial pseudogenes in the nuclear genome, also known as nuclear integrations of mitochondrial DNA (NUMTs), confounded our estimates of mitochondrial transcription. Previous work has shown that NUMTs are, in the vast majority of cases, integrated into intergenic/intronic regions of the nuclear genome, and are unlikely to be transcribed (*Collura et al., 1996*; *Dayama et al., 2014*). Nevertheless, to directly assess the contribution of NUMT RNA to our estimates of mtDNA expression, we implemented two measures: (1) examination of expression of mitochondrial pseudogenes annotated in Gencode (*Harrow et al., 2012*), and (2) comparison of two methods (featureCounts [*Liao et al., 2014*] and RSEM [*Li and Dewey, 2011*]) for quantifying expression levels, which treat differently those reads mapping to more than one genomic region. Results from both analyses support the notion that NUMTs do not confound our estimates of mtDNA expression (Materials and methods, *Figure 1—figure supplement 1* and *Figure 1—figure supplement 2*).

Across all tissues, mtRNAs were highly transcribed. However, we observed substantial differences in the abundance of any given mitochondrial (MT) transcript from one tissue to the next (as evaluated using TPM) (*Figure 1—figure supplement 3*). For example, compared to other tissues, samples derived from the kidney (TCGA studies KICH, KIRC, and KIRP, see Materials and methods for TCGA abbreviations) had exceptionally high levels of mtRNAs, while samples from lung adenocarcinoma (LUAD) expressed comparatively low levels of mtRNAs. When comparing our results to a prior study of MT protein abundances from the mouse (*Pagliarini et al., 2008*), we found good agreement between protein and mtRNA levels with respect to their ordering across tissues, (i.e. mtRNA expression and protein levels of MT-ND6 were highest in kidney, at moderate levels in liver, colon, and stomach, and lowest in lung).

### Differential expression of mtRNA in tumors compared to normal tissue

In (*Reznik et al., 2016*), we observed widespread mtDNA copy number depletion in tumors compared to matched adjacent normal tissue in a number of solid tumor types. Here, we first set out to test if expression of mtRNAs in these tumor types also was correspondingly lower. To do so, we estimated the differential expression of mtRNA transcripts in tumor vs. normal tissues for each of 13 cancer types with adequate numbers of tumor and normal samples (*Figure 2*). The majority of cancer types have a tendency for lower levels of mtRNAs, and five cancer types (breast, esophageal, head and neck, kidney clear cell, and liver) have lower levels of all 13 protein-coding mtRNAs. Among the seven cancer types with statistically significant depletion of mtDNA content in our earlier report (*Reznik et al., 2016*), all seven had lower mtRNA levels in at least 4/13 genes, with no mtRNAs

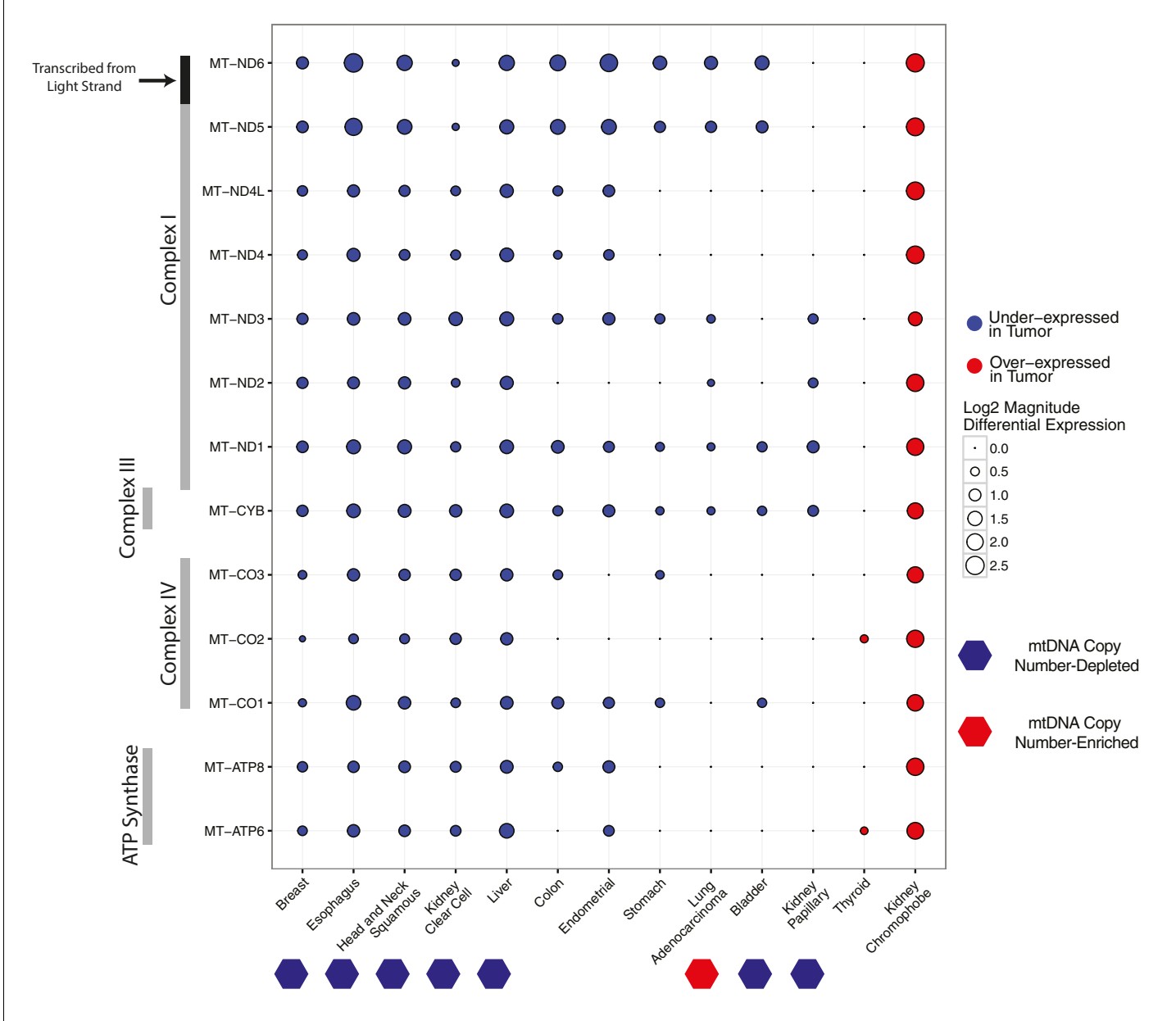

**Figure 2.** Differential expression of mitochondrial genes across cancer types. Magnitude and statistical significance of differential expression evaluated by limma voom (see Materials and methods). The majority of mtRNAs are strongly down-regulated in several cancer types, including esophageal, breast, head and neck squamous, kidney clear cell, and liver cancers. One cancer type (kidney chromophobe), shows increases in the abundance of mtRNAs. All tumor types showing mtDNA copy number depletion in tumors relative to adjacent normal tissue (bottom annotation) show analogous depletion of mtRNAs. In contrast, mtDNA copy number changes in lung adenocarcinomas and kidney chromophobes are not reflected in differential expression of mtRNAs.

showing over-expression in these tumor types. We also observed that LUAD (lung adenocarcinoma), which we found to be the only cancer type with increased mtDNA copy number in *Reznik et al. (2016)*, had lower expression of 6/13 mtRNAs, suggesting that any increase in mtDNA copy number in LUAD was compensated for at the transcriptional level. For one study, prostate cancer (PRAD), results were removed from analysis because differential expression using counts from featureCounts and RSEM produced differing results.

Breaking the general trend, one cancer type, KICH (chromophobe renal cell carcinomas), had higher levels of mtRNAs in tumors compared to normal tissues. The over-expression phenotype was notably large: each of the 13 genes had greater than 2-fold over-expression in tumors compared to normal tissue (*Figure 2*). Importantly, KICH did not have appreciable accumulation of mtDNA copies in our prior study.

The role of mitochondrial dysfunction, and of a mitochondrial accumulation phenotype, in chromophobe renal cell carcinomas has been appreciated for some time (*Nagy et al., 2002*). Two recent publications have highlighted the role that mtDNA mutations have in the development of one sub-stype of chromophobes, eosinophilic chromophobe renal cell carcinomas (*Davis et al., 2014*; *Joshi et al., 2015*). Eosinophilic KICH tumors have been proposed to arise from oncocytomas, a cancer type characterized by cytoplasms swollen with respiration-deficient mitochondria. This phenotype arises from two critical dysfunctions: somatic mtDNA mutations that render mitochondrial OXPHOS non-functional, and defective mitophagy that prevents clearance of dysfunctional mitochondria. Thus, it has been proposed that oncocytoma cells experience an energy crisis due to defective mitochondria/OXPHOS (perhaps sensed through AMPK), and respond by further upregulating the biogenesis of defective mitochondria (*Zong et al., 2016*). Thus, our observation of *increased* mtRNA levels in KICH tumors is likely to counterintuitively reflect a *drop* in respiration. As noted before, experimental respirometry measurements must be made to confirm this hypothesis.

Interestingly, we also found differences in the tendency for any one mtDNA-encoded gene to be differentially expressed across cancer types, which likely arises from the molecular details of mitochondrial transcription. The mitochondrial genome is transcribed in a polycistronic fashion, with all mRNAs and tRNAs on a strand transcribed simultaneously. Following transcription, tRNAs are excised from the transcript, and the majority of the remaining mRNAs are polyadenylated (*Pagliarini et al., 2008*; *Mercer et al., 2011*). Polycistronic transcription ensures that mtRNAs are highly co-expressed, although it is clear from many studies that mtRNAs undergo a large degree of post-transcriptional regulation, resulting in uneven steady-state abundances (*Rorbach and Minczuk, 2012*). With regard to differential expression, most of the genes encoding subunits of Complex I were downregulated in the majority of cancer types. In contrast, genes encoding subunits of Complex V (ATP6 and ATP8) and to a lesser extent Complex IV (MT-CO1,MT-CO2,MT-CO3) were generally under-expressed in only the strongly mtDNA- and mtRNA-depleted cancer types.

## Association of mtRNA with clinical parameters

We also evaluated the extent to which mtRNA levels were associated with clinical features (e.g. the age, pathological stage, and overall survival of patients), using available clinical data from the TCGA consortium (*Figure 3*, full results available in *Supplementary file 3*). Among these, papillary renal cell carcinoma (KIRP), esophageal carcinoma (ESCA), and thyroid cancer (THCA) showed an association between high mtRNA expression and increased age. It is not clear whether this statistical association is a secondary result of a correlation between age and other clinical/genomic features, (e.g. in THCA, age is positively associated with increased mutational density), and merits further investigation.

More interestingly, we identified five cancer types (ACC,KICH,LGG,PAAD, and LIHC) in which higher mtRNA expression levels were associated with increased overall survival. Three of these cancer types (ACC,KICH, and LGG), showed similar associations in our prior analysis using mtDNA copy number (i.e. high mtDNA copy number was associated with better overall survival (*Reznik et al., 2016*). KIRP tumors also showed an association between higher mtRNA expression and less aggressive disease, as assessed by pathological stage (*Supplementary file 3*). Interestingly, these results echo similar findings by reported by Gaude and Frezza, who reported an association between down-regulation of nuclear-DNA-encoded mitochondrial transcripts and poor clinical outcome across many cancer types (*Gaude and Frezza, 2016*).

## Correlation of mtDNA copy number and mRNA levels

Our simultaneous quantification of mtDNA copy number and mtRNA expression enabled us to address a more basic biological question: what is the relationship between the number of copies of mtDNA in a cell and the expression of mtDNA-encoded genes? A number of factors, including but not limited to mtDNA copy number, ultimately determine the steady-state abundance of mtRNAs

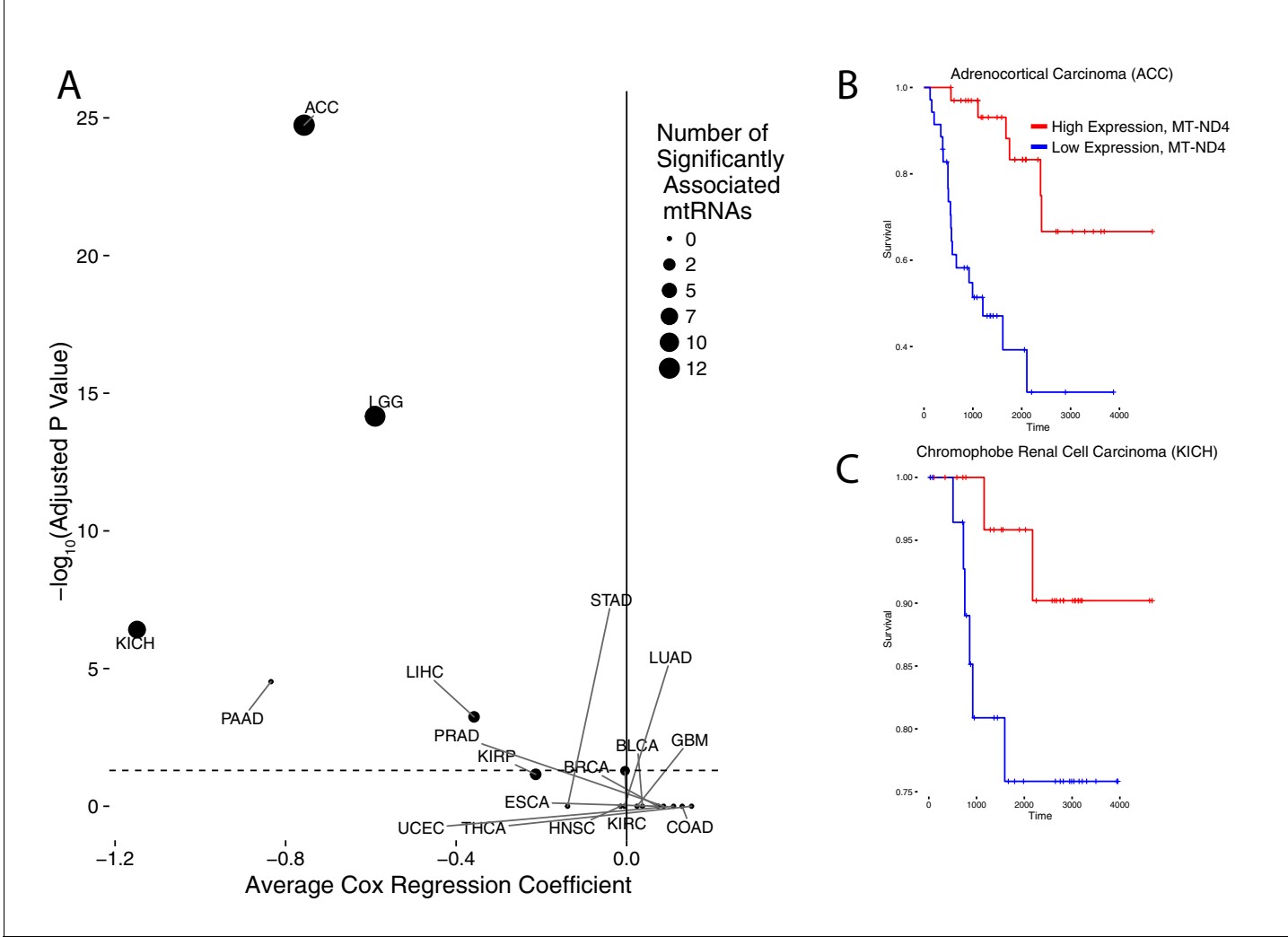

**Figure 3.** Association of mtRNA expression levels with overall survival across cancer types. Multiple-hypothesis-adjusted univariate p-values from Cox regression for each mtRNA are combined using Fisher's method for each cancer type. Several cancer types show an association between high mtRNA expression and improved outcome (negative Cox regression coefficient). Dashed line indicates threshold for statistical significance. Representative Kaplan-Meier plots are shown for overall survival in ACC (**B**) and KICH (**C**), partitioning patients into high expression vs. low expression groups based on median expression of MT-ND4.

and derived proteins in a cell. At low mtDNA copy number, transcript expression of mitochondrial genes may be limited by the number of DNA templates available for active transcription. Alternatively, other proteins (e.g. mitochondrial transcription termination factors) can control the rate of transcription, while yet others can modulate mtRNA stability and degradation (*Clemente et al., 2015*). Thus, it remains unclear whether mtDNA copy number is correlated to, and may be used as a surrogate for, the abundance of mtRNA.

To evaluate the association between mtDNA copy number and mtRNA abundance, we calculated (separately for each mtRNA) the non-parametric Spearman correlation between mtDNA copy number and mtRNA levels (in RSEM counts), for all samples with available mtDNA copy number and mtRNA expression estimates (18 tumor types analyzed in total, *Figure 4* and *Figure 4—figure supplement 1*). We found that the correlation between mtDNA copy number and mtRNA expression was highly tissue-type-specific. Seven of eighteen tumor types (adrenocortical, breast, glioma, kidney clear cell, kidney papillary, liver, and thyroid) had strong (BH-corrected p-value $<10^{-5}$) positive correlation across all or nearly all mtDNA-encoded genes (*Figure 4—figure supplement 1*). Seven

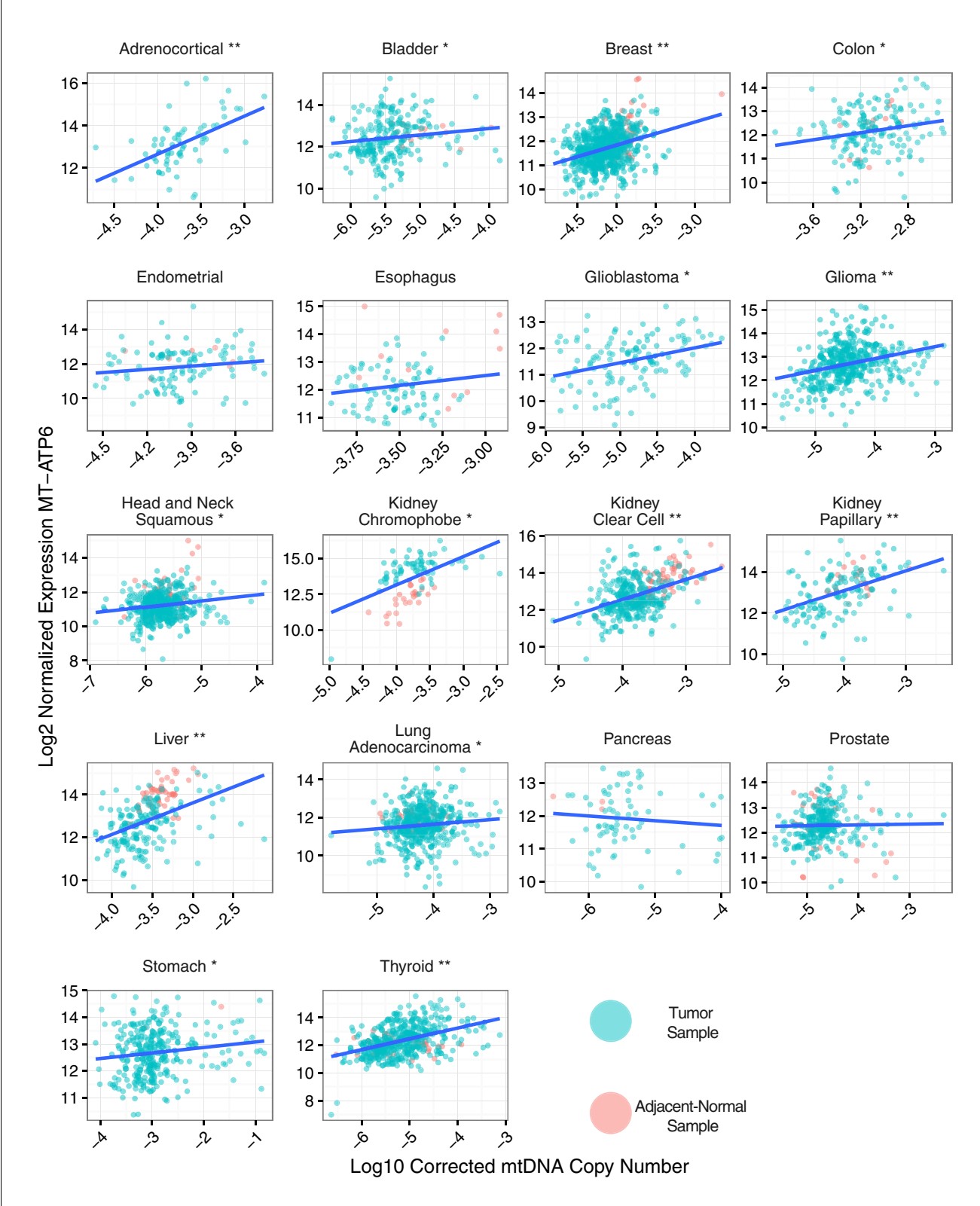

**Figure 4.** Correlation between mtDNA copy number and mtRNA expression. Relative mtDNA copy number was correlated against mtRNA expression (log$_2$ normalized counts from limma voom). The expression of one gene, MT-ATP6, is depicted, although other mtDNA protein coding genes are similar. Lines indicate best fit linear trend between mtDNA copy number and log$_2$ MT-ATP6 expression. Cancer types titled with an asterisk indicate a

*Figure 4 continued on next page*

*Figure 4 continued*

statistically significant correlation (Spearman, BH-adjusted p-value <0.05). Double asterisk indicates an especially strong correlation (BH-adjusted p-value $<10^{-5}$).

The following figure supplements are available for figure 4:

**Figure supplement 1.** Correlation between mtDNA copy number and mtRNA expression is highly dependent on cancer type, as well as on mtRNA gene.

**Figure supplement 2.** Correlation of RPPA with (**A**) mtDNA copy number (Spearman $\rho$ 0.18, p-value 0.003) and (**b**) MT-CO2 RNA expression (Spearman $\rho$0.38, p-value $<10^{-14}$) in KIRC.

**Figure supplement 3.** Correlation of mtDNA copy number and mtRNA expression in KICH.

cancer types (bladder, colon, glioblastoma, head and neck, kidney chromophobe, lung adenocarcinoma, and stomach) had weak but still statistically significant correlation. The remaining four cancer types (endometrial, esophagus, pancreatic, prostate) showed no statistically significant correlation. Of the seven cancer types with mtDNA depletion in tumors relative to normal tissue, five (all but esophageal and head and neck) showed positive correlation between mtDNA copy number and expression. Some genes (e.g. MT-ND5 and MT-ND6) had recurrently weaker correlation with mtDNA copy number, which may result from their lower (and thus potentially noisier) abundance compared to other MT transcripts. Thus, the data support a plausible correlation between mtDNA copy number and 'steady-state' mtRNA abundance across many but not all tissues.

For one cancer type (kidney clear cell), the TCGA consortium (*Creighton et al., 2013*) used reverse-phase protein arrays (RPPA) to measure the protein abundance of one mtDNA-encoded protein (MT-CO2) in tumors only (not normal tissue). This data enabled us to assess directly, in this one case, whether variation in mtDNA copy number and mtRNA levels translates to changes in protein levels. We found that MT-CO2 protein levels were strongly correlated with MT-CO2 mRNA (Spearman $\rho$ 0.38, p-value $<10^{-14}$), and to a lesser extent with mtDNA copy number (Spearman $\rho$ 0.18, p-value 0.003) (*Figure 4—figure supplement 2*).

Interestingly, correlations between mtDNA and mtRNA were not necessarily homogeneous between tumor and normal samples from the same tissue. Upon closer examination of samples from KICH (*Figure 4—figure supplement 3*), we found that tumor and normal samples had strong but distinct patterns of correlation between mtDNA copy number and mtRNA levels. In particular, because KICH tumors only had increased mtRNA levels compared to normals (and not an increase in mtDNA copy number), the best-fit trend line between mtDNA and mtRNA appears shifted vertically when comparing tumor to normal tissue samples. Put another way, it appears as if KICH tumors had higher values of the steady-state mtRNA/mtDNA ratio. Given that many KICH tumors harbor mtDNA mutations, these observations suggest that compensation for such mitochondrial dysfunction in KICH may be via transcriptional mechanisms, rather than changes to mtDNA ploidy.

## Patterns of nuOXPHOS and mtOXPHOS gene expression are not redundant

The mitochondrial genome encodes 13 polypeptides which serve as integral membrane subunits for 3 components of the electron transport chain and ATP synthase (herein referred to as mtOXPHOS proteins). The remaining ~80 OXPHOS subunits are encoded in the nuclear genome (herein referred to as nuOXPHOS proteins) (*Calvo and Mootha, 2010*). The protein levels of these subunits must be coordinated in order to maintain proper stoichiometry of OXPHOS complexes (*Cancer Genome Atlas Research Network et al., 2013*). Intriguingly, a study of the coordination of mtOXPHOS and nuOXPHOS protein levels in *S. cerevisae* found that the primary mode of regulation was through rapid translational, rather than transcriptional, synchronization (*Couvillion et al., 2016*). This leaves open the possibility that patterns of transcriptional changes in mtOXPHOS and nuOXPHOS may be asynchronous or even opposite.

To see if the nuclear and mitochondrial genome experience analogous changes in respiratory gene expression in the course of tumorigenesis, we examined the differential expression of

mtOXPHOS and nuOXPHOS subunits for all 13 cancer types with available expression data for both tumor and normal tissues (Figure 5). We calculated a differential expression (DE) score for each cancer type, which captured the tendency for a set of genes to be down-regulated (corresponding to a score of −1) or up-regulated (a score of +1) in tumors compared to normal tissue. In 9/13 cancer types, we observed agreement in DE scores between mtOXPHOS and nuOXPHOS subunits (e.g. see volcano plot of KIRC in (Figure 5). More interestingly, we found that four cancer types (BLCA, BRCA, LUAD, and UCEC) have opposite patterns of differential expression in mtOXPHOS and nuOXPHOS genes. For example, in BRCA, while nearly all mtOXPHOS genes are down-regulated in tumors compared to normal tissue (mirroring the mtDNA copy number depletion in BRCA), nearly all nuOXPHOS genes actually had increased mRNA levels.

For a functional electron transport chain, subunits of each OXPHOS complex must assemble with compatible stoichiometries in a process that requires coordination of individual subunit protein levels (*Mouchiroud et al., 2013*). Taken together, the above findings speak to the potential shortcomings of transcriptional data as surrogates for OXPHOS protein abundance and respiratory activity. As mentioned above, several recent studies have reported on the critical role of translational processes in the coordination of respiration. One report by Wagner, Kitami, and colleagues found that treatment of differentiated C2C12 murine myotubes with certain drugs (including eukaryotic translation inhibitors) induced anti-correlated changes in nuOXPHOS and mtOXPHOS genes, respectively (*Wagner et al., 2008*). Separately, Couvillion and colleagues reported that the sudden shift of *S. cerevisae* to a non-fermentable carbon source led to rapid induction of transcriptional changes in nuOXPHOS, but not mtOXPHOS. Instead, rapid redistribution of mitochondrial ribosomes supported fast adapation to respiration (e.g. via an increase in translation efficiency of Complexes III and IV). Together, these two studies suggest that the examination of translational activity (e.g. by ribosome profiling) may be useful tools for examining an imbalance in mtOXPHOS and nuOXPHOS transcription.

## Questions and future directions

The purpose of this study was to compare cancer-associated changes in mitochondrial transcript levels with changes in mtDNA copy number as reported in *Reznik et al. (2016)*. An equally important goal, for all scientific studies but perhaps more so for 'replication' studies of the sort here, is to lay out the loose ends and unanswered questions, and to make explicit those findings which did not agree with prior results. Below we offer a reflection on our findings, which may steer future investigations towards interesting territory for future discovery.

In tandem with our prior study (*Reznik et al., 2016*), this work finds that several tumor types suppress the expression of genes involved in oxidative phosphorylation/respiration. Five cancer types (kidney clear cell, kidney papillary, head and neck squamous cell, liver, and esophageal) were mtDNA-depleted, under-expressed mtRNAs, and under-expressed nuOXPHOS genes, in comparison to normal tissue. Importantly, because cells harbor excess respiratory capacity, a drop in the protein expression of mtDNA-encoded OXPHOS subunits will not necessarily precipitate a drop in respiration unless it is sufficiently large. Experimental measurements, e.g. respirometry or measurements of flux will be necessary to determine if drops in mtRNA expression translate to a drop in oxygen consumption/respiration (*Brand and Nicholls, 2011*). A reasonable subset of cancer types to initially investigate for such experiments would be the five listed above to suppress OXPHOS gene expression.

While changes in mtDNA copy number and mtRNA expression cannot be used as surrogates for changes in respiratory flux, several of the cancer types examined here are driven by genetic changes affecting mitochondrial respiration, suggesting that changes in mtRNA levels may reflect bona fide changes in respiration. For example, clear-cell renal cell carcinomas are driven by homozygous loss of VHL, which leads to activation of hypoxia inducible factor (HIF), and subsequently increased transcription of glycolytic enzymes (*Creighton et al., 2013*). The signature of HIF activation is evident not only in gene expression data, but also in metabolomic measurements of KIRC tumors (*Hakimi et al., 2016*), and suggests that KIRC tumors experience a drop in respiration. In particular, KIRC tumors exhibit increased levels (compared to normal tissue) of metabolites upstream of SDH (succinate dehydrogenase, Complex II), i.e. citrate, cis-aconitate, and succinate, and decreased levels of metabolites downstream of SDH (fumarate and malate). This partitioning of changes in metabolite levels could arise by a decrease in the activity of the electron transport chain and SDH, which would

induce a metabolic bottleneck and the accumulation (depletion) of metabolites upstream (downstream) of SDH.

Furthermore, while our data suggests that several cancer types may decrease mitochondrial respiration, mitochondrial metabolism provides several other important services to the cell. In some cases, respiration may be critical to these processes; for example, aspartate biosynthesis is heavily reliant on electron acceptors produced by mitochondrial respiration, and respiration-deficient cells become limited by the availability of aspartate (*Birsoy et al., 2015*). Beyond the ETC, cytosolic acetyl-CoA utilized for lipogenesis is produced from the action of ATP-citrate lyase on mitochondrial citrate, and substantial evidence supports the critical role of mitochondrial metabolism in producing adequate one-carbon precursors for nucleotide biosynthesis (*Ahn and Metallo, 2015*). Finally, it should be noted that xenografted $\rho^0$ (depleted of mtDNA) breast cancer cells form tumors only after re-acquiring mtDNA from the host, emphasizing that some basal level of mitochondrial respiration is likely required for tumor viability and malignancy (*Tan et al., 2015*).

When comparing the differential expression of mtRNA (tumor vs. adjacent-normal tissue) to changes in mtDNA copy number (tumor vs. adjacent-normal tissue), we found two cancer types (LUAD and KICH) had opposite changes in the two measures. To highlight a single example, lung adenocarcinomas were the only cancer type to accumulate mtDNA copies compared to normal tissue. Using mtRNA data, however, these tumors had lower mtRNA levels of 6/13 MT genes. We can speculate as to the source of this inconsistency: it may be that our inference from sequencing data of mtDNA copy number or mtRNA expression was erroneous for these tumor types. While this is a possibility, we would remind readers that the same analytical pipeline was applied to all cancer types, the majority of which had features consistent with our expectations, e.g. correlation of whole-exome and whole-genome estimates of mtDNA copy number in *Reznik et al. (2016)*, correlation of mtDNA copy number with expression of mitochondrially-localized genes in *Reznik et al. (2016)*, consistency of changes in mtDNA copy number and mtRNA expression across most cancer types in *Figure 2* of this study. Alternatively, if one takes our calculations to be reasonably accurate estimates of mtDNA and mtRNA abundance, then the unexpected results become more intriguing. Regarding the inconsistency between mtDNA ploidy and mtRNA abundance described above, it is known that a number of proteins regulate mtDNA transcription, including transcription factors (TFAM,TFB1M), transcriptional termination factors (MTERFs), as well as mitochondrial biogenesis in general (NRF-1 and NRF-2, PGC1$\alpha$,ERR$\alpha$) (*Scarpulla, 2008*; *Rorbach and Minczuk, 2012*). A more detailed investigation of these proteins may reveal the factors driving inconsistency between changes in mtDNA copy number and mtRNA.

The coordinated expression of mtDNA- and nuDNA-encoded subunits of the respiratory chain is critical to efficient respiration. Prior studies of mitochondrial-nuclear (mitonuclear) protein imbalance in *C. elegans* claimed that it promotes longevity via activation of the mitochondrial unfolded protein response (UPR$^{MT}$), a transcriptional response preserving mitochondrial function in the face of stress (*Cancer Genome Atlas Research Network et al., 2013*). Intriguingly, two recent studies reported activation of the UPR$^{MT}$ counterintuitively promoted tolerance of mutant mtDNA levels by activating mitochondrial biogenesis in an effort to recover OXPHOS activity (*Lin et al., 2016*; *Gitschlag et al., 2016*). If cancer cells indeed have substantial changes in mtDNA copy number and transcription (and mtDNA mutations), then activation of the mammalian UPR$^{MT}$ may be critical in supporting their proliferation in the face of increased mitochondrial stress.

Several of our findings discussed above allude to intriguing tissue-specific patterns of variation in mtDNA and mtRNA. Both the accumulation/depletion of mitochondrial transcripts (*Figure 2,4,5*) and their association with mtDNA ploidy were highly dependent on the cancer type. It is well-appreciated that the physiology and molecular constitution (*Fernández-Vizarra et al., 2011*; *Calvo and Mootha, 2010*) of mitochondria varies substantially from one tissue to the next. Can some of the unexpected phenomena described here (e.g. inconsistent changes in mtDNA copy number and mtRNA expression, decoupling of nuOXPHOS and mtOXPHOS) be explained by the tissue-specific physiology of mitochondria? We expect that, in the pursuit of such questions, tools for profiling the morphologic and proteomic composition of mitochondria, in parallel with organelle-targeted transcriptomics and genomics, will yield valuable insights.

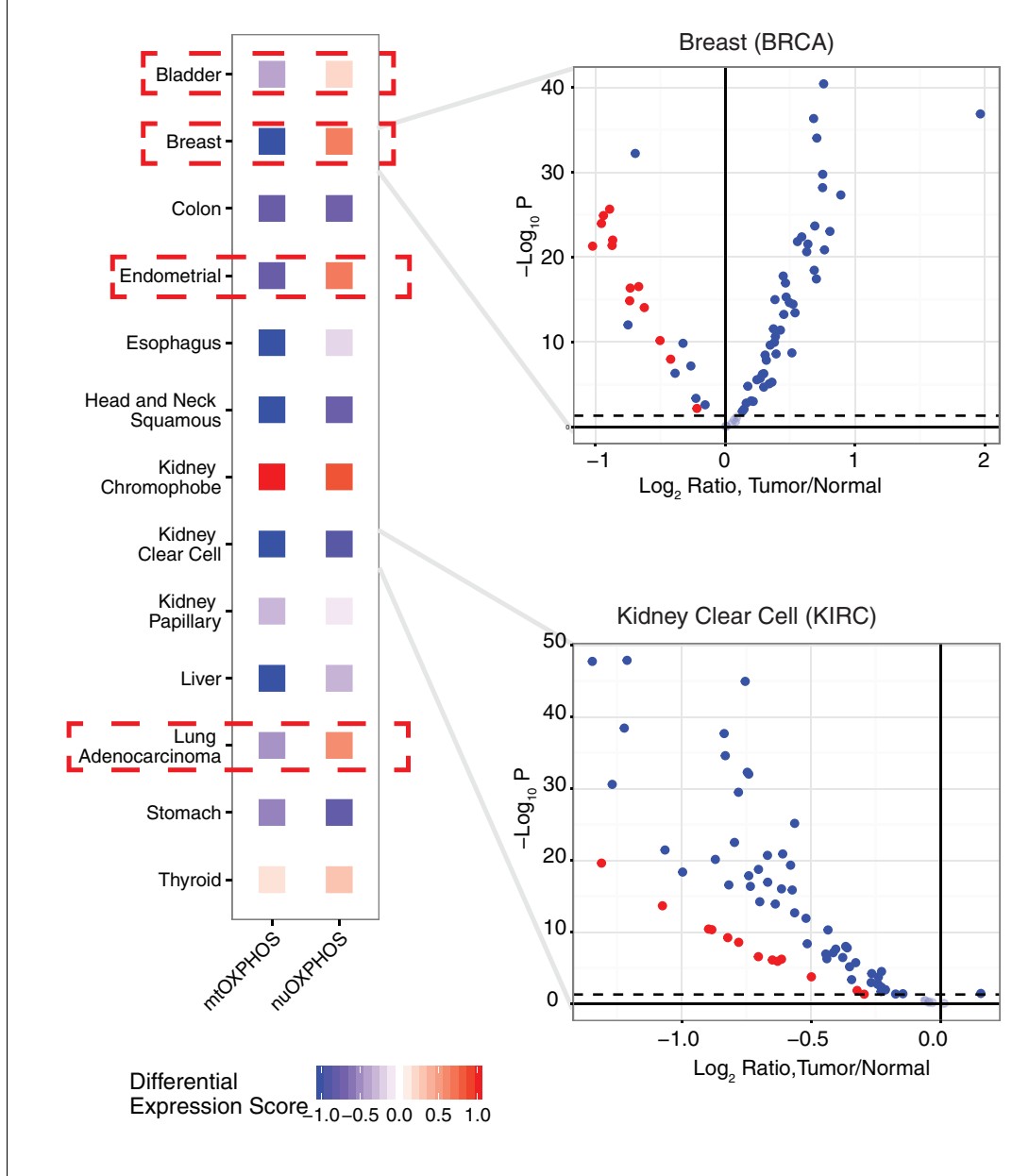

**Figure 5.** Comparison of differential expression (tumor vs. adjacent-normal tissue) of mtDNA-encoded OXPHOS subunits (mtOXPHOS) and nuclear-DNA-encoded OXPHOS subunits (nuOXPHOS). (A) Differential expression scores for mtOXPHOS and nuOXPHOS across cancers. Red dashed boxes highlight cancer types with opposite trends in mtOXPHOS and nuOXPHOS differential expression. (B) Volcano plots highlighting differential expression of mtOXPHOS (red) and nuOXPHOS (blue) genes in BRCA and KIRC.

## Materials and methods

### RNA sequencing alignment, quantification, and quality control

The input of our analysis pipeline is raw sequencing reads (in FASTQ format). The raw reads of the RNA-seq samples were retrieved from the Cancer Genomics Hub (CGHub, https://cghub.ucsc.edu, Research Resource Identifier RRID:SCR_003193). When FASTQ files were not available, e.g. for stomach adenocarcinoma, we downloaded aligned sequence reads (in BAM format) and extracted reads from BAM files with the Java program ubu.jar (https://github.com/mozack/ubu) before

processing samples using our pipeline. All acronyms for cancer types conform to the TCGA nomenclature (ACC: adrenocortical; BRCA:breast; BLCA: bladder; COAD: colon; ESCA: esophageal; GBM: glioblastoma; HNSC: head and neck squamous cell; KICH: kidney chromophobe; KIRC: kidney clear cell; KIRP: kidney papillary; LGG: low grade glioma; LIHC: liver, LUAD:lung adenocarcinoma; PAAD: pancreatic; PRAD: prostate; STAD: stomach; THCA: thyroid; UCEC: endometrial). Sample sizes for each cancer type are reported in *Supplementary file 1*.

We employed STAR aligner (*Dobin et al., 2013*), a fast accurate alignment software used widely in the NGS community, to map reads to UCSC human reference genome hg19 and reference transcriptome GENCODE (v19), using recommended parameters, e.g. '–outFilterType BySJout' and '–outFilterMultimapNmax 20', etc., which are also standard options of the ENCODE project for long RNA-seq pipeline. Samples with alignment rates less than 40% were excluded from further analysis.

The software tools RseQC (*Wang et al., 2012*) and mRIN (*Feng et al., 2015*) were used to evaluate sample quality. RNA degradation, as detected by mRIN (*Feng et al., 2015*), was present in some TCGA samples. Since degradation can bias expression level measurements and cause data misinterpretation, we excluded samples with evidence for degradation. Specifically, we used prostate cancer samples from the TCGA project as test data to decide on a degradation cutoff for mRIN. TCGA prostate samples had undergone extensive pathological, analytical, and quality control review and had been shown to include a significant portion of degraded samples (*Clemente et al., 2015*). We compared mRIN scores with RNA Integrity Numbers (RIN) calculated by TCGA for prostate samples and found they are highly negatively correlated (Pearson correlation $<-0.93$). To filter degraded samples, TCGA used a cutoff 7.0 for RIN, which corresponds roughly to $-0.11$ for mRIN. We also manually examined other tumor types including bladder urothelial carcinoma and breast invasive carcinoma and a mRIN cutoff $-0.11$ worked reasonably well for these studies as well. Hence, we used $-0.11$ as the degradation threshold for mRIN throughout our study. Samples with mRIN $<-0.11$ were regarded as degraded and, thus, excluded from further analysis.

When running STAR, we specified an option '–quantMode TranscriptomeSAM' to make STAR output a file, Aligned.toTranscriptome.out.bam, which contains alignments translated into transcript coordinates. This file was then used with RSEM (*Li and Dewey, 2011*) to quantify mitochondrial gene expression. The program 'rsem-calculate-expression' in the RSEM package requires strand specificity of the data, which is estimated using software RseQC (*Wang et al., 2012*).

To ensure that reads aligning to the mtDNA are not derived from NUMTs/mitochondrial pseudogenes in the nuclear DNA, we implemented two quality control measures. First, we examined the expression (using RSEM, which handles multi-mapping reads, see next paragraph) of 175 regions annotated as mtDNA pseudogenes in GENCODE v19. While these genes are a subsample of the total number of NUMTs in the genome (on the order of 1000 [*Dayama et al., 2014*]), they are nevertheless useful for roughly estimating the genome-wide expression of NUMTs. Across all studies, we failed to see substantial expression of NUMT transcripts. In general across all cancer types, the vast majority of samples showed 100- to 1000-folder greater expression of *bona fide* mtDNA-encoded genes than expression of mitochondrial pseudogenes (*Figure 1—figure supplement 1*).

As a second quality control measure, we compared estimates of expression counts produced from two computational approaches, RSEM and featureCounts. These methods differ fundamentally in the way that they count reads mapping to multiple regions in the genome (multi-mapping reads). RSEM handles multi-mapping reads using an expectation-maximization procedure, while featureCounts by default simply discards multi-mapping reads. Thus, if a read maps ambiguously to both mtDNA and a NUMT, then featureCounts will ignore this read while RSEM will not. For all 13 mtDNA-encoded mRNAs, we compared (1) the total expression (in counts) as reported by RSEM and featureCounts (*Figure 1—figure supplement 2A*) and (2) the tumor vs. normal differential expression estimate (*Figure 1—figure supplement 2B*). With the exception of a single study, we found excellent agreement between RSEM and featureCounts by both measures. In the PRAD study, we noted that the two approaches produced different results for differential expression of MT genes, and we therefore removed PRAD from our differential expression analysis. These data support the hypothesis, as reported by in prior work by others (*Dayama et al., 2014*; *Collura et al., 1996*), that NUMT expression is negligibly low and does not confound estimates of true mtDNA expression.

## Differential expression analysis

Differential expression analysis was completed using the limma voom package (*Law et al., 2014*). TMM normalization was applied to counts before running differential expression analysis. Estimates of differential expression were compared against analogous estimates using publicly available data from the Broad Institute's TCGA Firehose using the RTCGAToolbox R package (*Samur, 2014*), with excellent agreement between the two (data not shown). Cancer types with fewer than five adjacent-normal tissue samples profiled by RNA-Seq were discarded. As mentioned in the above section, the TCGA cancer type PRAD was removed from differential expression analysis due to differences when comparing results for mtDNA differential expression from RSEM and featureCounts.

Differential expression score is defined as

$$DE = \frac{\text{\# of Genes Overexpressed in Gene Set} - \text{\# of Genes Underexpressed in Gene Set}}{\text{\# of Genes in Gene Set}} \tag{1}$$

A score of 1 indicates that all genes in the gene set are overexpressed in tumor compared to normal tissue, while a score of $-1$ indicates that all genes in the gene set are underexpressed.

Genes included in the nuDNA OXPHOS geneset were derived from the MSigDB Reactome gene-set file (*Liberzon et al., 2015*).

## mtDNA copy number analysis

All data for mtDNA copy number analysis was derived from *Supplementary file 1* of (*Reznik et al., 2016*).

## Clinical association analysis

All data for clinical analysis was downloaded from the Broad Firehose. For survival calculations, Cox regression was used to associate $\log_2$ TPM expression of each mtRNA with overall survival. For association with pathological stage or age, non-parametric Spearman correlations were calculated. For each clinical variable, association was calculated with each of the 13 mtDNA-encoded mRNAs. Then, p-values for the associations were corrected using the Benjamini-Hochberg procedure, and then combined using Fisher's method. Because Fisher's method assumes independent p-values (which is not the case in this analysis, as mtRNAs are transcribed polycistronically), we additionally report in *Figure 3* the number of mtRNAs significantly associated (BH-adjusted p-value<0.05) with the clinical variable under examination.

# Additional information

## Funding

| Funder | Grant reference number | Author |
|---|---|---|
| National Cancer Institute | 5U24CA143840 | Ed Reznik<br>Nikolaus Schultz<br>Chris Sander |
| National Institutes of Health | P41GM103504 | Ed Reznik<br>Nikolaus Schultz<br>Chris Sander |
| Marie-Josee and Henry R. Kravis Center for Molecular Oncology | | Ed Reznik<br>Qingguo Wang<br>Konnor La<br>Nikolaus Schultz |
| National Cancer Institute | P30-CA008748 | Ed Reznik<br>Qingguo Wang<br>Konnor La<br>Nikolaus Schultz<br>Chris Sander |
| Roberston Foundation | | Nikolaus Schultz |
| Prostate Cancer Foundation | | Nikolaus Schultz |

The funders had no role in study design, data collection and interpretation, or the decision to submit the work for publication.

## Author contributions

ER, QW, NS, CS, Conception and design, Analysis and interpretation of data, Drafting or revising the article; KL, Analysis and interpretation of data, Drafting or revising the article

## Author ORCIDs

Ed Reznik, http://orcid.org/0000-0002-6511-5947

## Additional files

### Supplementary files

• Supplementary file 1. Summary of sample sizes for each cancer type.

• Supplementary file 2. Expression of 13 mtRNAs across all profiled samples.

• Supplementary file 3. Results of clinical analysis.

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
