## [Decision Letter]

[Editors’ note: this article was originally rejected after discussions between the reviewers, but the authors were invited to resubmit after an appeal against the decision.]

Thank you for submitting your work entitled "Mitochondrial Respiratory Gene Expression is Suppressed in Many Cancers" for consideration by *eLife*. Your article has been favorably evaluated by a Senior Editor and three reviewers, one of whom, Chi Van Dang, is a member of our Board of Reviewing Editors.

Our decision has been reached after consultation between the reviewers. Based on these discussions and the individual reviews below, we regret to inform you that your work will not be considered further for publication in *eLife*.

The manuscript by Reznik et al. reports alteration mitochondrial gene expression amongst human cancers, for which they have previously reported in *eLife* the spectrum of mtDNA alterations relative to normal tissues. Understanding somatic alterations of mitochondrial genome during tumorigenesis is critical for the next generation of cancer therapeutics. Several recent studies based on the TCGA specimens have revealed that mitochondria DNAs are altered in cancers, including mtDNA mutations as well as copy number changes. The authors previously reported that mtDNA copy number depletion in human cancers. In this follow-up study, they analyzed mitochondria gene expression using the TCGA RNA-seq profiles. They presented an expressional landscape of 13 mitochondria genes across >10 cancer types which have normal control specimens from the TCGA, and found that mtRNAs were dysregulated in cancers. This comprehensive mtRNA expression profile could contribute to better understanding both mitochondria biology and tumorigenesis.

However, there is a similar recent paper in Nature Communications on this topic by Frezza and colleagues "Tissue-specific and convergent metabolic transformation of cancer correlates with metastatic potential and patient survival".

Further, the mitochondrial genomic changes documented in this manuscript do not provide any insight into what advantage decreasing respiratory genes will have on tumorigenesis.

Complete loss of the respiratory chain impairs cancer cell proliferation and tumorigenesis. The authors do not observe a complete loss but a suppression of respiratory chain. What advantage does moderate suppression of respiratory chain provide for the cancer cells in vivo?

Additionally, two statements are not correct:

"Our results provide quantitative evidence for suppression of respiration across many cancer types." Any given cell only utilizes 10-30% of maximal respiratory capacity. Thus, a cell can have large drop in mtDNA without respiration being affected. The authors don't have any direct measurements that examine respiration.

"On the other hand, mtDNA depletion will not necessarily lead to a drop in respiration". If cells have complete mtDNA depletion then they will be respiratory deficient. mtDNA encodes for critical subunits of the respiratory chain.

Other omissions that detract from the paper include the following issues:

1) Is there clinical significance of mtRNA dysregulation in cancers? E.g. are the expression levels of mtRNAs correlated with disease stage/grade, patients' age, outcome and other clinical factors?

2) A detailed information for sample numbers of each cancer type needs to be provided, especially for the corresponding control specimens. Relatively small number of normal specimens may create biases for analyses.

---

## [Author Response]

[Editors’ note: the author responses to the first round of peer review follow.]

[…] However, there is a similar recent paper in Nature Communications on this topic by Frezza and colleagues "Tissue-specific and convergent metabolic transformation of cancer correlates with metastatic potential and patient survival".

The reviewers are correct in noting that a recent paper by Gaude and Frezza describes an association between aggressive disease and downregulation of mitochondrial gene expression across many cancers. While the Gaude paper complements our own, the two analyses are completely distinct. Gaude and Frezza use the expression of nuclear-DNA-encoded mitochondrial genes as a surrogate for respiratory gene expression. In contrast, our manuscript is the first (and to our knowledge the only one to date) to quantify and analyze expression of the mitochondrial genome. Publically available data from the TCGA consortium (used by Gaude and Frezza and in dozens of other manuscripts) does not quantify expression of mitochondrial DNA; mitochondrial genome expression data is in fact discarded prior to expression quantification.

We realized that we had not done a sufficiently good job in emphasizing the difference between other publications examining the expression of metabolic genes in cancer samples. We have created a new subsection and included additional text to clarify the novelty of our study with respect to prior work (“Quantifying Expression of mtRNAs Across Cancers). More importantly, because data for mtRNA expression is not available through the Broad Firehose data portal for TCGA data, we have included estimates of mtRNA expression across all 6,614 samples investigated as a supplementary resource with this publication ([Supplementary-material SD2-data]).

Furthermore, our analysis in Figure 5 demonstrates that expression of nuclear-DNA-encoded OXPHOS genes (used by Gaude and Frezza) is not redundant with mtDNA-encoded OXPHOS gene expression. Several cancer types (e.g. breast cancer, BRCA, and bladder cancer, BLCA) exhibit discordant changes in the expression of these sets of genes. These findings, and previous work by others examining mitonuclear coordination of gene expression, suggest that coupling between expression of the mitochondrial and nuclear genome, in addition to the level of gene expression itself, may become dysfunctional in these cancer types. Our data will be essential to those seeking to examine such phenomena.

*Further, the mitochondrial genomic changes documented in this manuscript do not provide any insight into what advantage decreasing respiratory genes will have on tumorigenesis.*

*Complete loss of the respiratory chain impairs cancer cell proliferation and tumorigenesis. The authors do not observe a complete loss but a suppression of respiratory chain. What advantage does moderate suppression of respiratory chain provide for the cancer cells* in vivo?

*Additionally, two statements are not correct:*

*"Our results provide quantitative evidence for suppression of respiration across many cancer types." Any given cell only utilizes 10-30% of maximal respiratory capacity. Thus, a cell can have large drop in mtDNA without respiration being affected. The authors don't have any direct measurements that examine respiration.*

*"On the other hand, mtDNA depletion will not necessarily lead to a drop in respiration". If cells have complete mtDNA depletion then they will be respiratory deficient. mtDNA encodes for critical subunits of the respiratory chain.*

The reviewer raises two critical and related points.

We agree that we did not adequately explain our results in the context of physiological mitochondrial respiration. Respirometry experiments can be used to estimate the difference between the basal oxygen consumption rate (OCR) of a cell and the maximal OCR upon decoupling of oxygen consumption from ATP synthesis (e.g. using an uncoupler such as FCCP). These kinds of experiments frequently (e.g. see Brand, Nicholls, Biochemical Journal 2011) demonstrate that cells harbor excess respiratory capacity. The implication, therefore, is that a reduction in functionally respiratory mitochondria in a tumor will not necessarily reduce the basal respiration rate of the tumor. However, in cases where the reduction in respiratory capacity is sufficiently large (e.g. large or complete mtDNA depletion), the basal respiratory rate of the cell will drop, because mtDNA encodes necessary components of the respiratory chain. These changes are now noted throughout the Introduction. We further emphasize in the discussion in subsection “Questions and Future Directions” that a drop in mtRNA expression does not necessarily indicate a drop in respiratory flux, but suggest that because of a consistent decrease in mtDNA copy number, mtOXPHOS expression, and nuOXPHOS expression in 5 cancer types, future experiments may initially focus on these cancers when investigating changes in respiration.

We further agree with the reviewer that a critical question remaining from our study is the selective advantage provided by a potentially moderate suppression of respiration. We are unable to perform in vivo or in vitro experiments to measure how suppression of mtRNAs affects respiratory flux and proliferation/malignancy. Indeed, an adequate investigation into the quantitative link between mtRNA levels and respiratory flux is a project in its own right. Nevertheless, we agree it is important to understand how potential changes in respiratory capacity enable tumors to survive selective pressures. To do so, we used three approaches.

In the first (Results section), we utilized established cancer biology to discuss how the observation of changes in mtRNA expression may arise from known tumorigenic driving mechanisms in two cancer types, clear-cell renal cell carcinoma (KIRC) and chromophobe renal cell carcinoma (KICH). In both of these cancer types, we propose that changes in mtRNA are an adaptive response to “driver” genetic mutations and upstream changes in cell signaling. For example, VHL inactivation in KIRC leads to activation of HIF which upregulates glycolytic genes and suppresses OXPHOS gene expression. The signature of HIF activation is present not only in gene expression data, but also in metabolomic data published by us (Hakimi et al. 2016, Cancer Cell), and suggests that KIRC tumors may experience a bona fide drop in respiration. In particular, KIRC tumors show a distinct metabolomic signature in TCA cycle metabolites, with those metabolites upstream of SDH (succinate dehydrogenase, Complex II), i.e. citrate, cis- aconitate, and succinate, exhibiting increased levels in tumors (compared to normal tissues) and those metabolites downstream of SDH exhibiting decrease levels. One explanation (but perhaps not the only) for this partitioning of changes in metabolite levels is a decrease in respiration and SDH activity, which would induce a metabolic bottleneck and the accumulation (depletion) of metabolites upstream (downstream) of SDH.

In the second approach (subsection “Association of mtRNA with Clinical Parameters”), we leverage the reviewer’s suggestion to discuss our results in light of available clinical data (Comment 1 below). We find that down-regulation of mtRNAs is significantly associated with decreased survival in 5 cancer types, including adrenocortical carcinomas and low grade gliomas. While this analysis does not speak to the mechanism by which changes in mtRNA expression are linked to a selective advantage, it does indicate that a drop in mtRNA expression is relevant, and is associated with more aggressive disease in some cancer types.

In the final approach, we felt it was also important to emphasize that mitochondria (and their metabolic “services”) are required for tumor cells to remain viable and malignant. In subsection “Questions and Future Directions, we briefly discuss the essential metabolic roles of mitochondria beyond respiration (e.g. production of biosynthetic precursors, production of aspartate, redox balance).

*Other omissions that detract from the paper include the following issues:*

*1) Is there clinical significance of mtRNA dysregulation in cancers? E.g. are the expression levels of mtRNAs correlated with disease stage/grade, patients' age, outcome and other clinical factors?*

We agree with the reviewer that, in light of the analysis in Gaude et al. (see our first response above), it would be interesting to analyze our results alongside clinical data. We have now analyzed our results with respect to clinical data provided by the TCGA, including age, pathological stage, and overall survival. In particular, we associated the expression of each mitochondrial mtRNA with overall survival (using Cox regression) and age and pathological stage (using non-parametric Spearman correlations). These results are described in Figure 3, subsection “Association of mtRNA with Clinical Parameters”, and in [Supplementary-material SD3-data], and are largely in line with clinical associations reported in our original study of mtDNA copy number across cancer. Overall, we find that several cancer types (e.g. adrenocortical carcinomas, low grade gliomas, chromophobe renal cell carcinomas) show an association between low expression of mtRNA and poor outcome. We also found several associations with age and stage, including an increase in mtRNA levels in older patients with thyroid tumors.

*2) A detailed information for sample numbers of each cancer type needs to be provided, especially for the corresponding control specimens. Relatively small number of normal specimens may create biases for analyses.*

This information is now available in [Supplementary-material SD1-data].